# Efficacy and acceptability of blue-wavelength light therapy for post-TBI behavioral symptoms: A systematic review and meta-analysis of randomized controlled trials

Karan Srisurapanont[1], Yanisa Samakarn[1], Boonyasit Kamklong[1], Phichayakan Siratrairat[1], Arina Bumiputra[1], Montita Jaikwang[1], Manit Srisurapanont[2]*

1 Faculty of Medicine, Chiang Mai University, Chiang Mai, Thailand, 2 Department of Psychiatry, Faculty of Medicine, Chiang Mai University, Chiang Mai, Thailand

* manit.s@cmu.ac.th

## Abstract

### Objective

Behavioral symptoms are common after traumatic brain injury (TBI), but their treatments remain unsatisfactory. This systematic review and meta-analysis compared the efficacy and acceptability between blue-wavelength light therapy (BWLT) and long-wavelength/no light therapy (LW/NLT) for post-TBI sleepiness, sleep disturbance, depressive symptoms, and fatigue.

### Methods

This study included randomized controlled trials comparing the effects of BWLT and LW/NLT on post-TBI sleepiness, sleep disturbance, depression, or fatigue. We searched Pubmed, Embase, CINAHL, and Cochrane Central Register of Controlled of Trials on April 13, 2022. The revised tool for assessing the risk of bias in randomized trials was applied. We performed a frequentist pairwise meta-analysis using a random-effects model.

### Results

Of 233 retrieved records, six trials (N = 278) were included in this meta-analysis. TBIs ranged from mild to severe, and the interventions were administered for a median of 35 days. Most trials delivered light therapy via lightboxes. Three trials had a high risk of bias. BWLT was significantly superior to LW/NLT in reducing sleep disturbance (5 trials; SMD = -0.63; 95% CI = -1.21 to -0.05; p = 0.03; $I^2$ = 61%) and depressive symptoms (4 trials; SMD = -1.00; 95% CI = -1.62 to -0.38; p < 0.01; $I^2$ = 56%). There were trends that BWLT was superior to LW/NLT in reducing sleepiness (6 trials; SMD = -0.92; 95% CI = -1.84 to 0.00; p = 0.05; $I^2$ = 88%) and fatigue (4 trials; SMD = -1.44; 95% CI = -2.95 to 0.08; p = 0.06; $I^2$ = 91%). All-cause dropout rates were not significantly different between groups.

**Data Availability Statement:** The data and r script of this study are available at https://doi.org/10. 6084/m9.figshare.19971236.v2.

**Funding:** This work was supported by a grant from Chiang Mai University, Chiang Mai, Thailand (grant no. 15/2565 for M.S.). However, the funders had no role in study design, data collection/analysis, decision to publish, or manuscript preparation.

**Competing interests:** The authors do not have any conflicts of interest to disclose.

## Conclusion

Limited and heterogenous evidence suggests that short-term BWLT is well accepted, has a large treatment effect on post-TBI depressive symptoms, and may have a moderate treatment effect on post-TBI sleep disturbance.

## Introduction

Traumatic brain injury (TBI) is increasingly recognized as a global health priority. In 2016, there were approximately 27·million new cases and 55 million people with a history of TBI, resulting in a loss of 8.1 million years of life living with disability [1]. Rates of cardiovascular, respiratory, musculoskeletal, digestive, urological, neurological, and psychiatric conditions are higher in the TBI population, leading to the increased use of health services [2]. TBI-related disability causes high healthcare costs and productivity losses, a large burden on individuals, families, healthcare systems, and economies [3].

Individuals with TBI are vulnerable to a broad range of mood, cognitive, and functional impairments. Three-quarters of these patients may have a psychiatric diagnosis in the first five years after TBI [4]. The prevalence rates of common behavioral symptoms following TBI are as follows: i) 30%-84% for sleep abnormalities, especially insomnia, circadian rhythm disorders, and sleepiness [5], ii) 27%-38% for clinically significant depression, major depressive disorder, or dysthymia [6], and iii) 18%-75% for fatigue [7]. Functional limitations and poor quality of life usually persist after developing post-TBI behavioral symptoms [8].

Treatment for post-TBI behavioral symptoms remains unsatisfactory. Only limited evidence suggested that aquatic physical activity, mindfulness-based stress reduction, and computerized working-memory training may be of benefit for post-TBI fatigue [9]. Although methylphenidate may decrease post-TBI depression [10], it can also cause anxiety, sleep difficulties, and increased blood pressure [11]. Antidepressants can impair cognitive function, in particular attention and memory, and may cause sleep disturbances [12].

Blue-wavelength light therapy (BWLT) may help reduce post-TBI behavioral symptoms. Light therapy may reverse the pathophysiology of depression by increasing serum serotonin levels and advancing the phases of sleep-wake cycles [13]. This therapy can reduce depressive symptoms in seasonal and non-seasonal affective disorders, as well as fatigue and sleepiness in seasonal affective disorder (SAD) [14–16]. Post-TBI depressive symptoms, fatigue, and sleep abnormalities are highly correlated [17], and this symptom cluster is relatively similar to the symptoms of SAD. Among all visible light wavelengths, blue/short wavelengths (λ = 400–490 nm) are the most potent synchronizing agent for altering physiological functions, particularly circadian rhythm [18, 19]. These lines of evidence suggest that blue-wavelength light therapy (BWLT) may be a treatment option for these behavioral symptoms after TBI.

There is a need for an updated review of light therapy for post-TBI behavioral symptoms. Two recent meta-analyses of treatments for post-TBI depression did not address the benefits of light therapy.[10, 20] While a recent network meta-analysis (NMA) of three trials (N = 117) found limited evidence suggesting the BWLT benefits for post-TBI depression and fatigue [21], another large trial (N = 131) was later published [22]. In this updated systematic review and meta-analysis, we proposed to compare the efficacy and acceptability between BWLT and long-wavelength/no light therapy (LW/NLT) for post-TBI sleepiness, sleep disturbance, depressive symptoms, and fatigue.

## Materials and methods

The protocol of this review was prospectively registered at Open Science Framework (https://osf.io/yf2qe/). The report of this systematic review was prepared based on the PRISMA 2020 Statements [23].

### Eligibility criteria and database searches

Inclusion criteria for a trial were as follows: i) randomized controlled trials, ii) adult patients (> 18 years old) with a history of TBI, iii) an experimental intervention of BWLT ($\lambda$ = 400–490 nm), iv) a control intervention of long-wavelength light therapy (LWLT, $\lambda$ > 490 nm) or no light therapy, and v) reporting the outcome of sleepiness, sleep disturbance, depression, or fatigue. To maintain the data consistency, we excluded the following trials: i) less than 50% of the participants having TBI, ii) light therapy not perceived by the eyes, and iii) concomitant treatment of biological or physical therapy.

We searched Pubmed, Embase, CINAHL, and Cochrane Central Register of Controlled of Trials on April 13, 2022. Four sets of search terms were intersected as follows: i) (brain injury) OR (head injury), ii) (light therapy) OR (light treatment) OR phototherapy, iii) sleep OR mood OR depress* OR fatigue OR exhaustion OR tiredness OR lethargy, and iv) random*. S1 Table in S1 Appendix shows the details of search strategies and results. There was no limitation on languages or publication years.

### Trial selection and data management

Records retrieved from database searches were imported into Zotero, a reference manager software, to merge the duplicates. After then, two reviewers (KS and YS) independently screened the titles and abstracts of unduplicated records, selected the items needing full-text articles, examined the full-text articles, selected the trials, and extracted the data using a data record form. Any discrepancies in these processes were dissolved by consensus discussion. References cited in the trial reports were manually searched. All trials meeting the eligible criteria were included in the systematic review and meta-analysis.

We collected the characteristics and data of each trial as follows: i) first author's family name, ii) publication year of the article, iii) country where the trial was conducted, iv) participants' characteristics, including TBI severity, mean age, prominent behavioral symptoms, the percentage of female participants, and mean duration after TBI, v) details of BWLT and LW/NLT, including the light therapy devices; vi) treatment duration and procedure, vi) the measures used to assess the outcomes, and vii) the trial results.

For each treatment arm, we collected the following results: i) the number of subjects being randomized, ii) the number of dropouts, iii) the number of subjects included in the data analysis, and iv) baseline, final, and change mean scores (SDs) of four outcomes (i.e., sleepiness, sleep disturbance, depression severity, and fatigue measures). If an outcome was assessed at several time points, we collected only the end-of-treatment data. The follow-up data after treatment discontinuation were disregarded.

We extracted the data from published articles and, if available, added those from the reports of registered clinical trials. Graph data were estimated using the WebPlotDigitizer, version 4.5 [24].

### Risks of bias assessment

Two reviewers (KS and YS) independently assessed the risk of bias in the included trials using the RoB2—a revised Cochrane risk-of-bias tool for randomized trials [25]. The RoB2 appraised

five domains of bias as follows: i) randomization process, ii) adherence to the assigned intervention(s), iii) missing outcome data, iv) measurement bias, and v) the bias of the reported results. Each domain could be rated as low risk, some concerns, and high risk. The bias level of the least favorable domain was used as the overall risk of bias in each trial.

## Effect measures and synthesis methods

We calculated the standardized mean differences (SMDs) of the change in behavioral scores because these outcomes were usually assessed by various scales. The dropout rates were compared between groups using risk ratios (RRs).

This meta-analysis combined the data from participants given LWLT and those receiving NLT as a single control group (the LW/NLT group). This decision was made because previous RCT and NMA found that long-wavelength light therapy (LWLT) ($\lambda > 490$ nm, e.g., amber, red) and no light therapy (NLT) had no different treatment effects on post-TBI sleepiness, sleep disturbance, depressive symptoms, and fatigue [21, 26]. Participants receiving BWLT ($\lambda = 400$–490 nm) were considered the experimental group.

We pooled the SMDs using a frequentist random-effects meta-analysis and the restricted maximum-likelihood estimator for heterogeneous variance. The pooled SMDs (Hedges' g) were calculated using the inverse variance-weighted average method. Because some trials might have zero dropouts in both arms, we pooled the RRs of dropout rates using the Mantel-Haenszel method with the continuity correction of 0.5 for zero cells [27]. We also computed the z-statistics to determine whether the average effect size differed significantly from zero. A *p*-value of z-statistic less than 0.05 indicated that the treatment effects of BWLT and LW/NLT were significantly different. The SMDs of 0.2, 0.5, and 0.8 were considered small, medium, and large effect sizes, respectively [28, 29].

For the trials that reported only the standard errors or the 95% CI of change scores, we converted these values to the change-score SDs using RevMan (version 5.4), Cochrane Collaboration [30]. For the missing SDs of change mean scores, we calculated a correlation coefficient (CE) from the trials reporting baseline, final, and change SDs of their corresponding mean scores. After then, we applied that CE to impute the missing SDs of change mean scores as recommended by the Cochrane Handbook for Systematic Reviews of Interventions [31].

Forest plots were used to display the trial data, the pooled results, and the data heterogeneity. The $I^2$ values between 50% - 75% and $> 75\%$ were considered substantial and considerable heterogeneity, respectively [31, 32]. We conducted subgroup analyses to assess the moderating effects of TBI severity on the treatment effects. The sensitivity analyses were performed to explore how conclusions might be affected if the trials with a high risk of bias were excluded. The small-study effect of each outcome was visualized using a funnel plot. If ten trials or more were included in the meta-analysis, we would quantify the funnel plot asymmetry using the Egger's test [33].

We used the *imputeSD* function of the *MKInfer* package (version 0.6) to impute the missing SDs [34]. The data were analyzed and visualized using the *Meta* package (version 5.2.0) [35]. All analyses were performed in the *R* Program (version 4.2), R Foundation for Statistical Computing [36].

## Assessing the certainty of evidence

Two reviewers (KS and YS) independently applied the GRADE approach to rate the certainty of meta-analysis results [37]. Evidence obtained from randomized controlled trials was initially rated as high-quality evidence. The evidence quality would be downgraded by one or two levels if there was any concern as follows: i) risk of bias, ii) inconsistency, iii) indirectness, iv)

imprecision, and v) publication bias. The evidence quality could be rated up by one or two levels if the result showed a large effect size, a dose-response gradient, or no effect of plausible residual confounding. The overall judgment on each meta-analysis outcome was finally classified as high, moderate, low, or very low certainty.

# Results

## Trial selection and characteristics

Database searches found 233 records, which were 159 non-duplicated records. After the title and abstract screening, we reviewed 14 full-text documents and finally included six trials (N = 278) (see Fig 1) [22, 26, 38–41]. The data from a registered trial report were added to

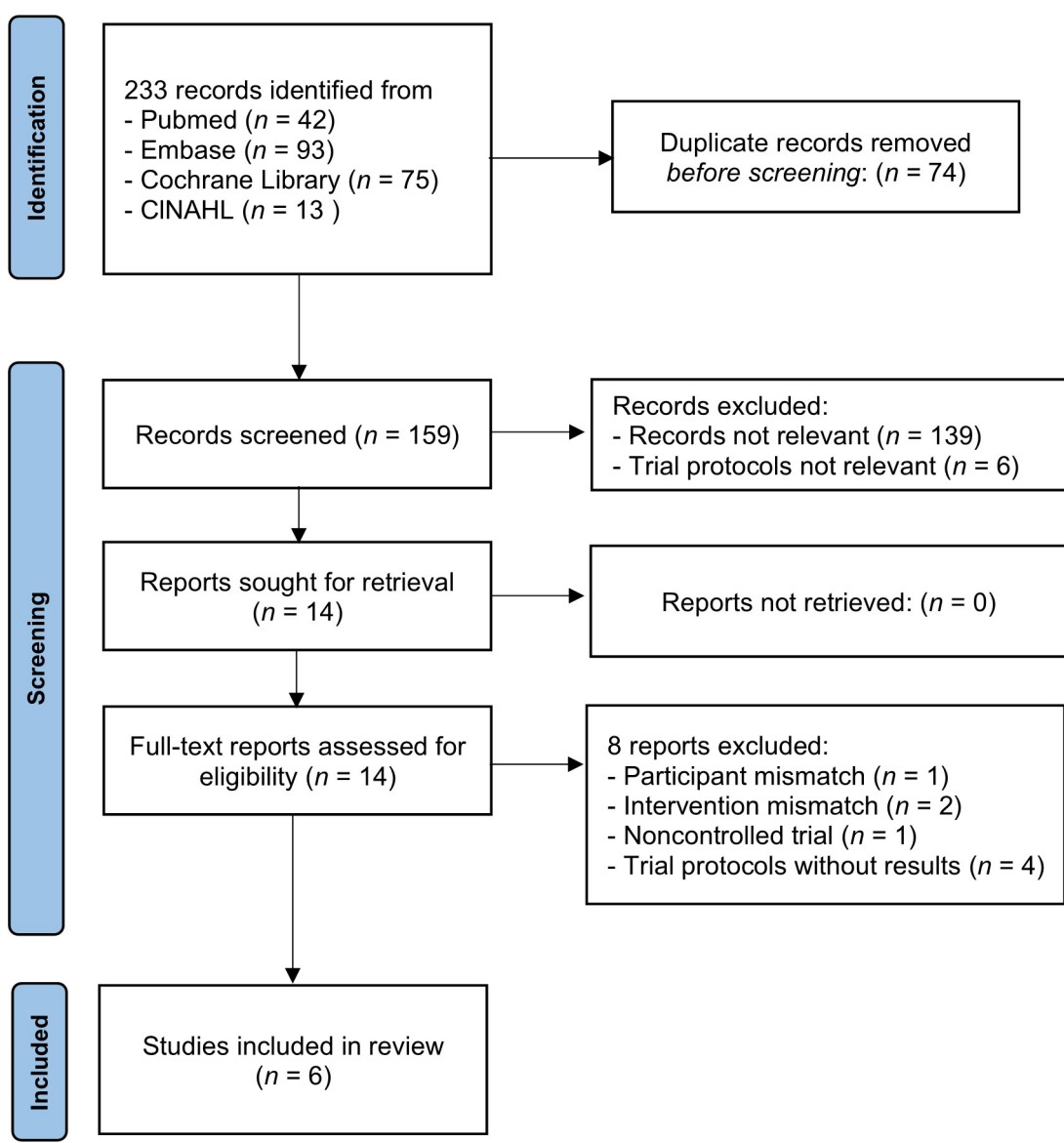

**Fig 1. PRISMA diagram showing the process of trial selection.** The systematic review and meta-analysis searched four databases, removed the duplicated records, and screened the titles and abstracts of records. After examining 14 full-text reports of relevant records, six trials were quantitatively synthesized.

those of a published article [39, 42]. Eight reports were excluded due to the participant mismatch (n = 1), intervention mismatch (n = 2), no control arm (n = 1), and trial protocols without results (n = 4) (see S2 Table in S1 Appendix).

Among the 278 participants in six included trials, 135 and 143 received BWLT and LWLT/NLT, respectively. Of the six trials, one had three arms of BWLT (n = 10), LWLT (ambient light therapy) (n = 10), and NLT (n = 10) [26]. The data of LWLT and NLT groups of this three-arm trial were combined as a control group. The other five trials had two arms comparing BWLT and LWLT/NLT. The severity levels of TBI were mild (2 trials), mild to severe (2 trials), moderate to severe (1 trial) [22], and severe (1 trial) [40] (see Table 1). Medians (interquartile range, IQR) of the means of age, the percentages of female participants, and treatment duration were 38.8 (28.6–41.7) years, 43.3% (34.5% - 51.1%), and 35 (28–42) days, respectively. Devices used to administer light therapy were lightboxes (4 trials) [22, 26, 39, 41], face-mounted device resembling glasses (1 trial) [40], and house lighting (1 trial) [38].

## Trial results and risk of bias in the included trials

As reported by the included trials, BWLT reduced the following symptoms more significantly than LW/NLT: i) sleepiness in 3 of 6 trials, ii) sleep disturbance in 3 of 5 trials, iii) depressive symptoms in 2 of 3 trials, and iv) fatigue in 2 of 4 trials (see Table 1). No trial compared the dropout rates between groups statistically.

Because the participants' awareness of NLT would affect outcome assessment, two trials with an NLT arm were judged to have a high risk of bias in this respect (see Fig 2) [26, 40]. In addition, the awareness in these two trials might also affect the participants' adherence to treatments, which resulted in a risk of bias on deviations from intended interventions. A randomized cross-over trial had a high risk of bias in the randomization process [38]. Based on these reasons, the overall risk of bias was low in three trials and high in the other three trials.

## Meta-analysis results

The pooled data revealed that BWLT was significantly superior to LW/NLT in reducing sleep disturbance (5 trials; SMD = -0.63; 95% CI = -1.21 to -0.05; p = 0.03; $I^2$ = 61%) and depressive symptoms (4 trials; SMD = -1.00; 95% CI = -1.62 to -0.38; p < 0.01; $I^2$ = 56%) (see Fig 3). Despite not statistically significant, there were trends that BWLT was superior to LW/NLT in reducing sleepiness (6 trials; SMD = -0.92; 95% CI = -1.84 to 0.00; p = 0.05; $I^2$ = 88%) and fatigue (4 trials; SMD = -1.44; 95% CI = -2.95 to 0.08; p = 0.06; $I^2$ = 91%). The pooled dropout rates showed no significantly different between groups (6 trials; RR = 0.66; 95% CI = 0.33 to 1.32; p = 0.24; $I^2$ = 0%) (see S1 Fig in S1 Appendix).

Subgroup analyses based on TBI severity levels revealed that BWLT reduced sleepiness in patients with mild TBI to a significantly greater extent than those with mild to severe TBI ($\chi 2$ = 9.13, df = 1, p < 0.01) (see Table 2 and S2 Fig in S1 Appendix). BWLT was not significantly different between subgroups in reducing sleep disturbance ($\chi 2$ = 0.00, df = 1, p = 0.97) and depressive symptoms ($\chi 2$ = 0.44, df = 1, p = 0.51). No subgroup analysis was performed for the fatigue outcome because the data of this outcome were drawn from the participants with mild to severe TBI only. Dropout rates were also not significantly different between subgroups ($\chi 2$ = 0., df = 1, p = 0.52).

The sensitivity analyses were performed by excluding three trials with a high risk of bias [26, 38, 40] (see Table 2 and S3 Fig in S1 Appendix). After the trial exclusion, BWLT showed no significantly superiority of BWLT for sleepiness disturbance (3 trials; SMD = -1.39; 95% CI = -2.76 to -0.03; p = 0.05, $I^2$ = 93%). The analysis also reversed the significant superiority of BWLT to a nonsignificant one for both sleep disturbance (2 trials; SMD = -0.60; 95% CI =

**Table 1. Characteristics of trials included in the systematic review and meta-analysis[*,†].**

| Study (country, study) | Participants | | | | | | Treatment | | | | Outcomes |
|---|---|---|---|---|---|---|---|---|---|---|---|
| | TBI level | Prominent symptoms | % Female | Mean age (years) | Initial GCS | Mean duration after TBI | Experimental | Control | Administration | Treatment duration (days) | |
| Sinclair 2014 (Australia) | Mild to severe | sleepiness, sleep disturbance, or fatigue | 20.0 | 42.0 | N/A | 36.9 months | BWLT ($\lambda_{max}$ = 465 nm) (n = 10) | ALT ($\lambda_{max}$ = 574 nm) or NLT (n = 20) | Lightbox for 45 minutes daily in the morning | 28 | • Sleepiness: ESS (Experimental > Control) • Sleep disturbance: PSQI (Experimental ≈ Control) • Depressive symptoms: BDI-II (Experimental ≈ Control) • Fatigue: FSS (Experimental > Control) • Dropout rate: (N/A) |
| Quera Salva 2020 (France) | Severe | sleepiness, sleep disturbance, or fatigue | 45.0 | 36.6 | 5.94 | 9.03 years | BWLT ($\lambda_{max}$ = 468 nm) (n = 10) | NLT (n = 10): | Face-mounted device resembling glasses for 30 minutes daily after waking | 28 | • Sleepiness: ESS (Experimental ≈ Control) • Sleep disturbance: PSQI (Experimental ≈ Control) • Depressive symptoms: HRSD-17 (Experimental > Control) • Fatigue: FSS (Experimental > Control) • Dropout rate: (N/A) |
| Killgore 2020 (USA) | Mild | sleepiness and sleep disturbance | 53.1 | 23.3 | N/A | 6.75 months | BWLT ($\lambda_{max}$ = 469 nm) (n = 16) | ALT ($\lambda_{max}$ = 578 nm) (n = 16) | Lightbox for 30 minutes daily in the morning. | 42 | • Sleepiness: ESS (Experimental > Control) • Sleep disturbance: PSQI (Experimental > Control) • Depressive symptoms: BDI-II (N/A) • Dropout rate: (N/A) |
| Raikes 2020 (USA) | Mild | sleep disturbance | 62.8 | 25.9 | N/A | 9.20 months | BWLT ($\lambda_{max}$ = 480 nm) (n = 17) | ALT ($\lambda_{max}$ = 530 nm) (n = 18) | lightbox for 30 minutes daily in the morning | 42 | • Sleepiness: ESS (Experimental > Control) • Sleep disturbance: PSQI (Experimental > Control) • Depressive symptoms: BDI-I: (Experimental > Control) • Dropout rate: (N/A) |
| Bell 2021 (USA) | Moderate to severe | sleep disturbance | 31.0 | 40.9 | 3.12 | 29 days | BWLT ($\lambda_{max}$ = 440–480 nm) (n = 65) | RLT ($\lambda_{max}$ > 650 nm) (n = 66) | lightbox for 30 minutes daily in the morning | 10 | • Sleepiness: KSS (Experimental ≈ Control) • Fatigue: BNI-FS (Experimental ≈ Control) • Dropout rates (N/A) |
| Connolly 2021 (Australia) | Mild to severe | fatigue | 41.7 | 44.3 | N/A | 10.2 years | BWLT (n = 17) | Sham control condition (n = 13) | House-based light therapy installed in the room that the participants spent most time | 60 | • Sleepiness: ESS (Experimental ≈ Control) • Sleep disturbance: PSQI (Experimental > Control) • - Fatigue: BFI (Experimental ≈ Control |

[*] The symbols of > and ≈ indicate significant superiority (p < 0.05) and no significant difference (p ≥ 0.05) as being reported by the authors, respectively.

[†] N/A indicates no statistical analysis of the difference between groups.

ALT: Amber Light Therapy; BWLT: Blue-Wavelength Light Therapy; NLT: No Light Therapy; RLT: Red Light Therapy.

BDI-I: Beck Depression Inventory; BDI-II: Beck Depression Inventory, 2nd edition; BFI: Brief Fatigue Inventory; BNI-FS: Barrow Neurological Institute Fatigue Scale; ESS: Epworth Sleepiness Scale; FSS: Fatigue Severity Scale; GCS: Glasgow Coma Scale; HRSD-17: 17-item Hamilton Rating Scale for Depression; KSS: Karolinska Sleepiness Scale; PSQI: Pittsburgh Sleep Quality Index; TBI: Traumatic brain injury.

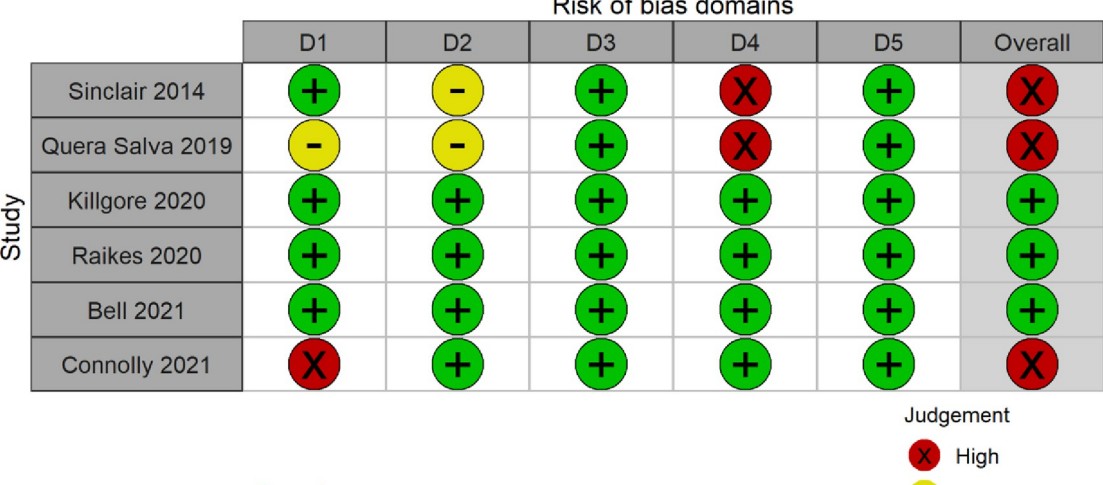

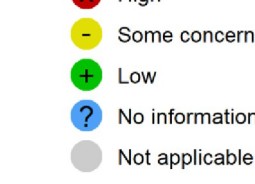

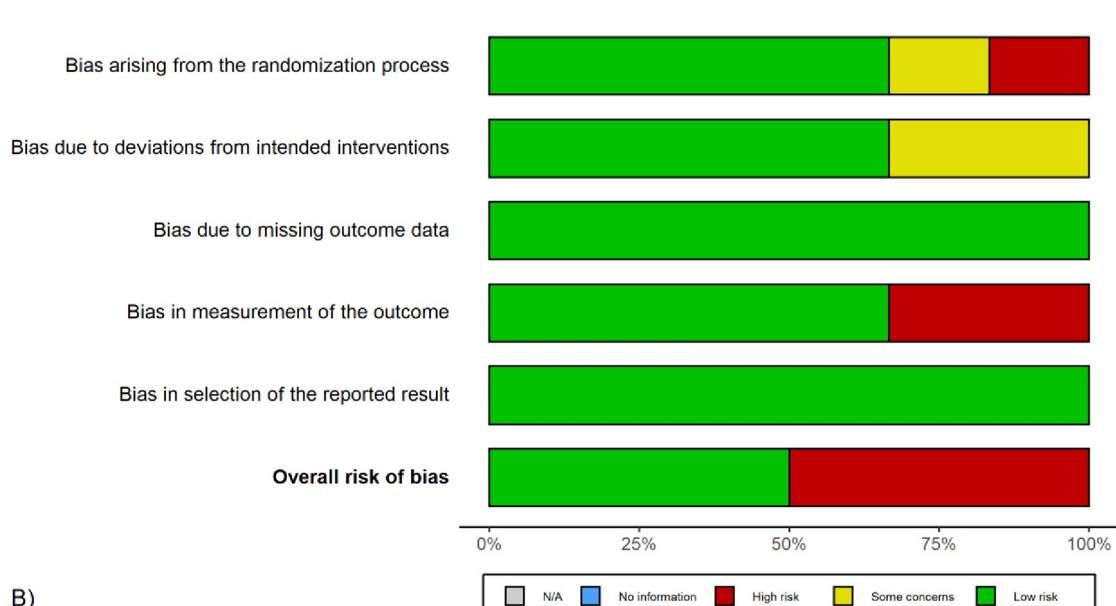

**Fig 2. Risk of bias in the included trials.** A) Risk of bias in each domain of individual trials and B) Risk of bias across all included trials. Risk of bias were assessed using version 2 of the Cochrane risk-of-bias tool for randomized trials (RoB 2).

-2.27 to 1.07; p = 0.48; $I^2$ = 89%) and depressive symptoms (2 trials; SMD = -0.79; 95% CI = -2.00 to 0.41; p = 0.20; $I^2$ = 80%). The nonsignificant effects of BWLT on fatigue (1 trial; SMD = -0.13; 95% CI = -0.49 to 0.22) and dropout rates (3 trials; RR = 0.75; 95% CI = 0.35 to 1.65; p = 0.49; $I^2$ = 1%) still remained.

Fig 4 and S4 Fig in S1 Appendix show the funnel plots of five outcomes. The funnel plots for sleepiness and fatigue outcomes were subjectively asymmetry, indicated by the far apart

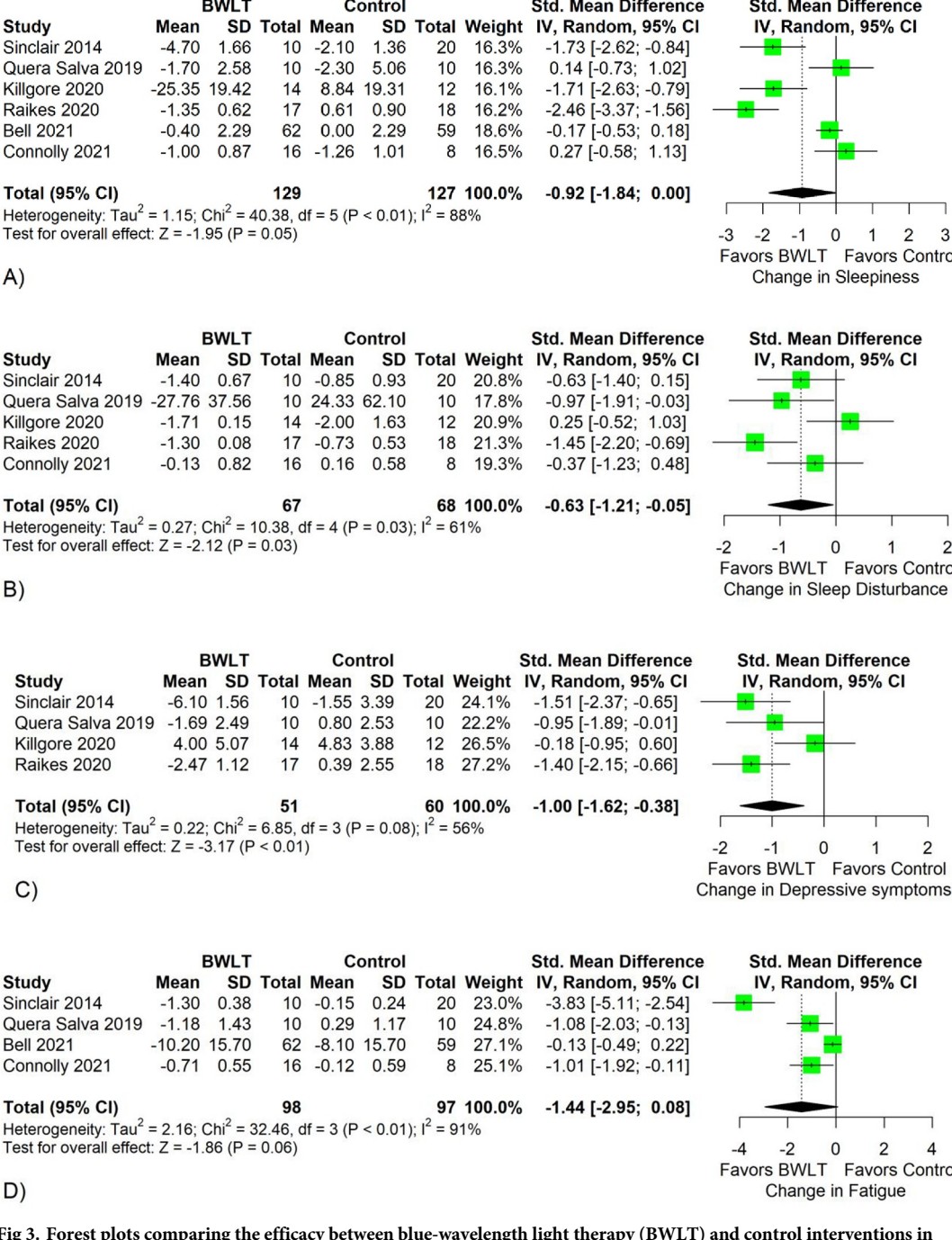

**Fig 3. Forest plots comparing the efficacy between blue-wavelength light therapy (BWLT) and control interventions in reducing behavioral symptoms.** Behavioral symptoms included sleepiness (A), sleep disturbance (B), depressive symptoms (C), and fatigue (D). Standardized mean differences (SMDs) expressed the differences in depressive symptom reduction between groups. The diamonds indicate the treatment estimates of the total sample. Control interventions included amber, red, or no light therapy.

lines displaying treatment estimates and no treatment effect. The funnel plots for sleep disturbance, depression, and dropout rates were relatively symmetry. Due to the small number of included trials, no Egger's test was performed.

**Table 2. Results of subgroup and sensitivity analyses.**

| Outcome | Results (n) | Heterogeneity |
|---|---|---|
| **Subgroup analysis** | | |
| • Sleepiness | $\geq$ Mild (n = 4): SMD (95% CI) = -0.35 (-1.21; 0.50) | $I^2$ = 77% |
| | Mild (n = 2): SMD (95% CI) = -2.08 (-2.84; -1.33) | $I^2$ = 28% |
| | Subgroup difference: $\chi 2$–9.13, df = 1, p < 0.01 | |
| • Sleep disturbance | $\geq$ Mild (n = 3): SMD (95% CI) = -0.64 (-1.13; -0.15) | $I^2$ = 0% |
| | Mild (n = 2): SMD (95% CI) = -0.60 (-2.27; 1.07) | $I^2$ = 89% |
| | Subgroup difference: $\chi 2$–0.00, df = 1, p = 0.97 | |
| • Depressive symptoms | $\geq$ Mild (n = 2): SMD (95% CI) = -1.25 (-1.89; -0.62) | $I^2$ = 0% |
| | Mild (n = 2): SMD (95% CI) = -0.79 (-2.00; 0.41) | $I^2$ = 80% |
| | Subgroup difference: $\chi 2$–0.44, df = 1, p = 0.51 | |
| • Fatigue | $\geq$ Mild (n = 4): SMD (95% CI) = -1.44 (-2.95; 0.08) | $I^2$ = 91% |
| | (No trials in mild TBI) | |
| • Dropout rates | $\geq$ Mild (n = 4): RR (95% CI) = 0.56 (0.24; 1.30) | $I^2$ = 0% |
| | Mild (n = 2): RR (95% CI) = 1.07 (0.18; 6.39) | $I^2$ = 0% |
| | Subgroup difference: $\chi 2$ = 0.41, df = 1, p = 0.52 | |
| **Sensitivity analysis** | | |
| • Sleepiness | n = 3: SMD (95% CI) = -1.39 (-2.76; -0.03), p for z-score = 0.05 | $I^2$ = 93% |
| • Sleep disturbance | n = 2: SMD (95% CI) = -0.60 (-2.27; 1.07), p for z-score = 0.48 | $I^2$ = 89% |
| • Depressive symptoms | n = 2: SMD (95% CI) = -0.79 (-2.00; 0.41), p for z-score = 0.20 | $I^2$ = 80% |
| • Fatigue | n = 1: SMD (95% CI) = -0.13 (-0.40; 0.22), p for z-score = N/A | $I^2$ = N/A |
| • Dropout rates | n = 3: RR (95% CI) = 0.76 (0.35; 1.65), p for z-score = 0.49 | $I^2$ = 1% |

SMD: standardized mean difference; RR: risk ratio; TBI: traumatic brain injury.

## Certainty of evidence

Table 3 summarizes the evidence certainty and the rationales used to rate the certainty of BWLT effects. Four domains of the treatment effects were neutral and had no rating up or down. Those included: i) high initial certainty of evidence (RCTs), ii) no concern on indirectness (all participants had a history of TBI), iii) no dose-response of treatment effect, and iv) no evidence showing the exclusion of all plausible residual confounding. The risk of bias was high across all outcomes. The issues of inconsistency, imprecision, publication bias, and large effect were varied among the outcomes. Based on the above rationales, the final certainty levels of BWLT effects were judged as follows: i) moderate for depressive symptoms, ii) low for sleep disturbance and dropout rates, and iii) very low for sleepiness and fatigue.

## Discussion

Limited and heterogenous evidence suggests that short-term BWLT is well accepted and may be effective for some behavioral symptoms commonly found in patients with TBI. While moderate certainty evidence supports the large treatment effect of BWLT in reducing depressive symptoms, only low certainty evidence suggests the moderate treatment effect of BWLT in relieving sleep disturbance. Preliminary evidence reveals that sleepiness after mild TBI may better respond to BWLT than that after severe TBI.

This study was an updated meta-analysis of a previous one [21]. Apart from the larger sample size and the number of trials, as the rationales mentioned earlier, this update performed a PMA instead of NMA. The present PMA found that BWLT was effective for depressive symptoms. Among the four trials included in this study, only one found no superiority of BWLT

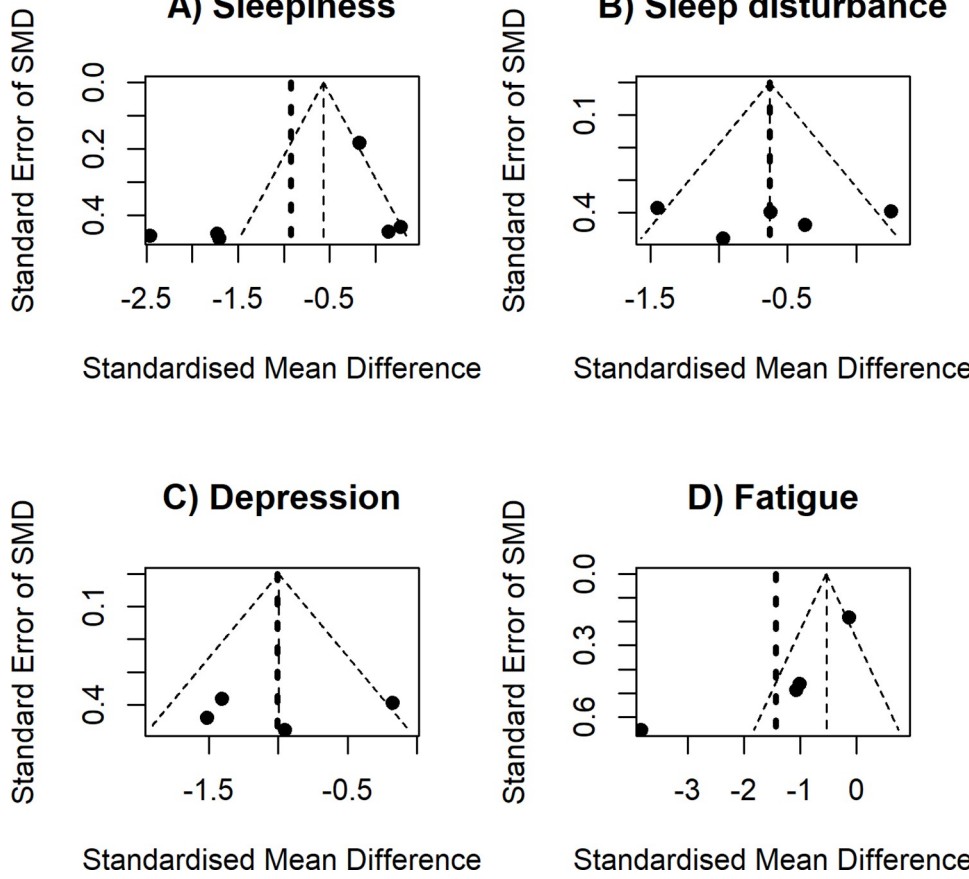

**Fig 4. Funnel plot showing the relationships between behavioral symptoms reduced by BWLT and their precision.**
Standardized mean differences (SMDs) of behavioral symptom reduction indicate the effect estimates of Blue-
Wavelength Light Therapy (BWLT) compared with control interventions. The standard errors of those SMDs express
the precision. The outer dashed lines indicate the triangular region within which 95% of studies are expected to lie in
the absence of both biases and heterogeneity. The bold dash line corresponds to no intervention effect. Control
interventions included amber, red, or no light therapy.

over LW/NLT in this respect [26]. Although this negative result could not be explained, it
should be noted that the participants in this trial were relatively older (mean age of 42.0 years)
than those in the other three trials (mean age ranged between 25.9 and 36.6 years). As shown
by reduced melatonin suppression, elderly individuals appear to respond to blue light exposure
less than the young group [43]. It was possible that this negative finding might reflect the age-
related changes in lens density.

This PMA found the BWLT benefit for sleep disturbance and a trend of the BWLT benefit
for sleepiness, which were not revealed in the previous NMA [21]. Our more favorable results
might be caused by the addition of positive results from Connelly's and Bell's trials, respec-
tively [22, 38]. The finding that BWLT could relieve sleepiness is in line with previous findings
showing that light exposure is associated with significant improvement in subjective and
objective alertness [44].

The statistically significant benefit of BWLT for fatigue found in the previous NMA turned
out to be a beneficial trend in this PMA [21]. This dissimilarity might relate to the least favor-
able results of two recent trials [22, 38].

TBI can cause various brain dysfunctions, including circadian abnormalities. Circadian
rhythm is regulated primarily in the suprachiasmatic nucleus (SCN) of the anterior

**Table 3. The certainty of BLWT treatment effects for four behavioral symptoms.**

| Outcomes | Initial certainty of evidence | Lower if | | | | | Higher if | | | Overall certainty of evidence |
|---|---|---|---|---|---|---|---|---|---|---|
| | | Risk of bias | Inconsistency | Indirectness | Imprecision | Publication bias | Large effect | Dose response | All plausible residual confounding | |
| Sleepiness | High (RCTs) | -1 (high risk in 3 of 6 trials) | - 2 ($I^2$ = 88%) | 0 (all participants had a history of TBI) | -1 (p = 0.05) | -1 (asymmetry plot) | +1 (SMD = 0.92) | 0 (no evidence) | 0 (no evidence) | Very low |
| Sleep disturbance | | -1 (high risk in 3 of 5 trials) | - 1 ($I^2$ = 61%) | | 0 (p = 0.03) | 0 (no asymmetry plot) | 0 (SMD = 0.63) | | | Low |
| Depressive symptoms | | -1 (high risk in 2 of 4 trials) | - 1 ($I^2$ = 56%) | | 0 (p < 0.01) | 0 (no asymmetry plot) | +1 (SMD = 1.00) | | | Moderate |
| Fatigue | | -1 (high risk in 2 of 4 trials) | - 2 ($I^2$ = 91%) | | -1 (p = 0.06) | -1 (asymmetry plot) | +1 (SMD = 1.44) | | | Very low |
| Dropouts | | -1 (high risk in 3 of 6 trials) | 0 ($I^2$ = 0%) | | -1 (p = 0.24) | 0 (no asymmetry plot) | 0 (RR = 0.66) | | | Low |

BWLT: Blue-wavelength light therapy; RCT: randomized controlled trial; SMD = standardized mean differences; RR = risk ratio.

hypothalamus and can affect sleep, wakefulness, and all physiologic functions that vary across the day, including melatonin secretion [45]. TBI-induced circadian disruption may be caused by many brain functions, e.g., circadian hormone regulation and neurotransmitter functions [46]. Neuroinflammation due to brain injury may also initiate a prolonged immune response that changes the expression of clock genes in the SCN and circadian function [47, 48]. Clinical findings suggest two common types of post-TBI circadian rhythm disorders, including delayed sleep phase syndrome and irregular sleep-wake patterns [49]. These rhythm disorders are consistent with the attenuated and delayed melatonin releases at night [50]. Besides sleep-wake cycle abnormalities, several lines of evidence also suggest that circadian disturbance plays a role in the pathophysiology of depression [51]. Fatigue appears to be a symptom following depression and/or sleep disturbances.

Light exposure during the daytime, especially BWLT administered in the morning, may resynchronize the disrupted circadian rhythm and relieve circadian-related behavioral symptoms. Light regulates the sleep-wake cycle by sending the signals from intrinsically photosensitive retinal ganglion cells (ipRGCs), photoreceptors in the mammalian eyes, to SCN, lateral hypothalamus, ventrolateral preoptic nucleus, intergeniculate leaflet, and supraparaventricular nucleus [46]. A similar process via the amygdala also affects mood processing. Although light affects many pathways in the brain, those on SCN neurons may best explain its benefit on the sleep-wake cycle and mood disorders. SCN neurons adjust their circadian rhythm according to the input of ambient light and humoral/autonomic nervous system signals to and from the rest of the body. While nighttime light can delay the circadian clock, daytime light can advance this clock [52]. Among all lights, blue light is the most potent synchronizing agent for the circadian system [19]. Therefore, providing BWLT in the morning or daytime would be able to advance the circadian phase, which is a reverse process of the delayed circadian phase due to TBI. After the restoration of circadian rhythm, post-TBI behavioral symptoms should be

resolved. Serum serotonin levels increased by BWLT may also play a role in reducing post-TBI depressive symptoms [13].

This meta-analysis included the largest number of RCTs examining the benefits of BWLT for post-TBI behavioral symptoms. While the previous results were mixed and inconclusive, the results of this meta-analysis seemed to clarify at least two points: i) BWLT had a large treatment effect in reducing depressive symptoms (moderate certainty), and ii) BWLT had a moderate treatment effect in relieving sleep disturbance (low certainty).

The evidence included in this systematic review and meta-analysis had some limitations. First, this review had a small sample size of 278 patients with TBI participating in six trials. Second, the available data are highly heterogeneous, possibly due to various study designs, e.g., participants' age and TBI severity, light therapy devices, control interventions, and outcome measures. These two limitations are of concern for some study outcomes. For example, this study found large treatment effects of BWLT for sleepiness (SMD = -0.92) and fatigue (SMD = -1.44), but they did not show statistically significant superiority. These inconclusive results might be caused by the combination of small sample size and data heterogeneity. Third, the study duration of the included trials was relatively short (a median of 35 days). While post-TBI behavioral symptoms are chronic conditions, long-term outcomes of BWLT have not been known.

Our review process also had a limitation. Due to the small number of included trials, we combined the participants given sham light therapy (placebo) and those receiving no light therapy (no treatment) as a single control group. Evidence suggests that placebo and no treatment are not the same. Trials using placebos in mentally-ill patients often have lower estimated effects of the experimental intervention than those using no-treatment controls [53].

Our findings suggest that BWLT should be a treatment option for common behavioral symptoms after TBI. Although BWLT is safe and may help relieve post-TBI depression and sleep disturbance, its common adverse events should be monitored, e.g., nausea, diarrhea, headache, and eye irritation [54]. While short wavelengths near 460 nm are very efficient in shifting the circadian phase, intensive exposure to short-wavelength light between 400 and 440 nm should be avoided. Evidence suggests that these wavelengths can damage the retina via a photochemical reaction called photoreversal of bleaching [18].

The present results should be helpful for clinical practice and research in this area. Because treatment options for post-TBI behavioral symptoms remain unsatisfactory, as a promising intervention, BWLT may change the clinical practice and prognosis of patients with post-TBI depression and sleep disturbance. Clinicians may consider BWLT as a treatment option for post-TBI behavioral symptoms. In a recent review [55], guidelines suggested only neurobehavioral interventions and cognitive-behavioral therapy for rehabilitating behavioral and emotional disorders. The low to moderate certainty of our positive findings may impact future evidence-based practice guidelines and lead to a recommendation of BWLT for some post-TBI behavioral symptoms, e.g., depression, sleep disturbance. More trials in large sample sizes with longer study duration remain needed. Two reasons appear to support that patients with mild TBI should be a priority studied group. First, other than the benefits for depression and sleep disturbance, our subgroup analysis also found that BWLT might improve sleepiness in patients with mild TBI. Second, little has been known about the specific therapy for behavioral symptoms in mild TBI [55].

## Conclusions

Limited and heterogenous evidence suggests that short-term BWLT is well accepted, has a large treatment effect on post-TBI depressive symptoms, and may have a moderate treatment

effect on post-TBI sleep disturbance. Future trials in large sample sizes with longer study duration are warranted.

## Supporting information

**S1 Checklist.**
(DOCX)

**S1 Appendix.**
(PDF)

## Acknowledgments

**Systematic review protocol:** https://osf.io/yf2qe/ (Open Science Framework).

## Author Contributions

**Conceptualization:** Karan Srisurapanont, Yanisa Samakarn, Boonyasit Kamklong, Phichayakan Siratrairat, Arina Bumiputra, Montita Jaikwang, Manit Srisurapanont.

**Data curation:** Karan Srisurapanont, Yanisa Samakarn, Boonyasit Kamklong, Phichayakan Siratrairat, Manit Srisurapanont.

**Formal analysis:** Karan Srisurapanont, Manit Srisurapanont.

**Investigation:** Karan Srisurapanont, Yanisa Samakarn, Boonyasit Kamklong, Phichayakan Siratrairat, Manit Srisurapanont.

**Methodology:** Karan Srisurapanont, Yanisa Samakarn, Boonyasit Kamklong, Phichayakan Siratrairat, Arina Bumiputra, Montita Jaikwang, Manit Srisurapanont.

**Validation:** Karan Srisurapanont, Yanisa Samakarn, Boonyasit Kamklong, Manit Srisurapanont.

**Visualization:** Karan Srisurapanont, Yanisa Samakarn, Manit Srisurapanont.

**Writing – original draft:** Karan Srisurapanont, Yanisa Samakarn, Boonyasit Kamklong, Manit Srisurapanont.

**Writing – review & editing:** Karan Srisurapanont, Yanisa Samakarn, Boonyasit Kamklong, Manit Srisurapanont.

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
