## [Decision Letter · Decision Letter 0]

22 Jul 2022

PONE-D-22-16609Efficacy and acceptability of blue-wavelength light therapy for post-TBI behavioral symptoms: a systematic review and meta-analysis of randomized controlled trialsPLOS ONE

Dear Dr. Srisurapanont,

Thank you for submitting your manuscript to PLOS ONE. After careful consideration, we feel that it has merit but does not fully meet PLOS ONE’s publication criteria as it currently stands. Therefore, we invite you to submit a revised version of the manuscript that addresses the points raised during the review process.

We look forward to receiving your revised manuscript.

Kind regards,

Tariq Jamal Siddiqi

Academic Editor

PLOS ONE

Journal Requirements:

The authors do not have any conflicts of interest to disclose.

Reviewers' comments:

Reviewer's Responses to Questions

**Comments to the Author**

1. Is the manuscript technically sound, and do the data support the conclusions?

Reviewer #1: Yes

2. Has the statistical analysis been performed appropriately and rigorously? 

Reviewer #1: Yes

3. Have the authors made all data underlying the findings in their manuscript fully available?

Reviewer #1: Yes

4. Is the manuscript presented in an intelligible fashion and written in standard English?

Reviewer #1: Yes

5. Review Comments to the Author

Reviewer #1: The authors have conducted a meta-analysis on, "Efficacy and acceptability of blue-wavelength light therapy for post-TBI behavioral symptoms: a systematic review and meta-analysis of randomized controlled trials". In my opinion, the manuscript can be improved by incorporating the following points:

1. The authors have mentioned a very detailed inclusion criteria. However, the manuscript lacks any statement regarding the exclusion criteria

2. More information should be mentioned about WEBPLOTDIGITIZER. Moreover, adding a reference would be ideal.

3. Can the authors add the company names of the software used in the meta-analysis ?

4. It is necessary to cite which trials reported administering BWLT via light box or glass or house lighting.

5. In my opinion, the authors should emphasize on the strengths of this article. The authors need to highlight what are the gaps in the literature that required this study, and that how the results of this review help with filling the gaps in the medical literature.

Regarding the "Table 1" :-

Overall, the manuscript is very well written and I appreciate that the authors have presented accurate results. However, please note that the table in its current form is very confusing. The table should have proper sections for control/intervention and sample/age/bmi/duration/female or male percentage/characteristics/outcomes.

6. PLOS authors have the option to publish the peer review history of their article (what does this mean?). If published, this will include your full peer review and any attached files.

Reviewer #1: No

---

## [Author Response · Author response to Decision Letter 0]

27 Jul 2022

Thank you for the valued comments given by the reviewer. They are very much helpful in improving our manuscript. We have revised the manuscript accordingly and wish to give details on each point of revision in the following statements. In the mark-up copy, the revision points are in red. The pages and paragraphs are also referred to those in the mark-up copy of the manuscript. Please be informed that the cleaned manuscript is for press and has no mark-up inside.

1. The authors have mentioned a very detailed inclusion criteria. However, the manuscript lacks any statement regarding the exclusion criteria

• Our response: We have added the following sentence.

• Page 5, Paragraph 1: To maintain the data consistency, we excluded the following trials: i) less than 50% of the participants having TBI, ii) light therapy not perceived by the eyes, and iii) concomitant treatment of biological or physical therapy.

2. More information should be mentioned about WebPlotDigitizer. Moreover, adding a reference would be ideal.

• Our response: We have added the version and reference #24 of the WebPlotDigitizer.

• Page 6, Paragraph 3: We extracted the data from published articles and,…WebPlotDigitizer, version 4.5 [24].

• Reference #24: Rohatgi A. WebPlotDigitizer. Pacifica, California, USA; 2011. Available: https://automeris.io/WebPlotDigitizer

3. Can the authors add the company names of the software used in the meta-analysis ?

• Our response: We have added the company names of the software used in the meta-analysis.

• Page 8, Paragraph 3: All analyses were performed in the R Program (version 4.2), R Foundation for Statistical Computing [36].

4. It is necessary to cite which trials reported administering BWLT via light box or glass or house lighting.

• Our response: We have added citations after each device.

• Page 9, Last paragraph & Page 10, Paragraph 1: Devices used to administer light therapy were lightboxes (4 trials) [22,26,39,41], face-mounted device resembling glasses (1 trial) [40], and house lighting (1 trial) [38].

5. In my opinion, the authors should emphasize on the strengths of this article. The authors need to highlight what are the gaps in the literature that required this study, and that how the results of this review help with filling the gaps in the medical literature.

• Our response: We have added a paragraph to emphasize the strength of this article.

• Page 20, Paragraph 2: This meta-analysis included the largest number of RCTs examining the benefits of BWLT for post-TBI behavioral symptoms. While the previous results were mixed and inconclusive, the results of this meta-analysis seemed to clarify at least two points: i) BWLT had a large treatment effect in reducing depressive symptoms (moderate certainty), and ii) BWLT had a moderate traetment effect in relieving sleep disturbance (low certainty). These results should be helpful for research and treatment planning in this area.

6. Regarding the "Table 1" :-

Overall, the manuscript is very well written and I appreciate that the authors have presented accurate results. However, please note that the table in its current form is very confusing. The table should have proper sections for control/intervention and sample/age/bmi/duration/female or male percentage/characteristics/outcomes.

• Our response: We have extensively revised Table 1 accordingly to the reviewer’s comment

• Page 11-12: Please see Table 1.

Remark

Please be informed that we have added three sentences to inform about the trials with two and three arms that included in this study

Page 9, Paragraph 3: Of the six trials, one had three arms of BWLT (n = 10), LWLT (ambient light therapy) (n = 10), and NLT (n = 10) [26]. The data of LWLT and NLT groups of this three-arm trial were combined as a control group. The other five trials had two arms comparing BWLT and LWLT/NLT.

---

## [Decision Letter · Decision Letter 1]

16 Aug 2022

PONE-D-22-16609R1Efficacy and acceptability of blue-wavelength light therapy for post-TBI behavioral symptoms: a systematic review and meta-analysis of randomized controlled trialsPLOS ONE

Dear Dr. Srisurapanont,

Thank you for submitting your manuscript to PLOS ONE. After careful consideration, we feel that it has merit but does not fully meet PLOS ONE’s publication criteria as it currently stands. Therefore, we invite you to submit a revised version of the manuscript that addresses the points raised during the review process.

We look forward to receiving your revised manuscript.

Kind regards,

Tariq Jamal Siddiqi

Academic Editor

PLOS ONE

Journal Requirements:

Reviewers' comments:

Reviewer's Responses to Questions

**Comments to the Author**

1. If the authors have adequately addressed your comments raised in a previous round of review and you feel that this manuscript is now acceptable for publication, you may indicate that here to bypass the “Comments to the Author” section, enter your conflict of interest statement in the “Confidential to Editor” section, and submit your "Accept" recommendation.

Reviewer #1: All comments have been addressed

2. Is the manuscript technically sound, and do the data support the conclusions?

Reviewer #1: Yes

3. Has the statistical analysis been performed appropriately and rigorously? 

Reviewer #1: Yes

4. Have the authors made all data underlying the findings in their manuscript fully available?

Reviewer #1: Yes

5. Is the manuscript presented in an intelligible fashion and written in standard English?

Reviewer #1: Yes

6. Review Comments to the Author

Reviewer #1: I appreciate and thank the authors for their significant contribution and for submitting a revised draft. All the previous comments have been addressed.

However, my concerns remain regarding the strengths of the findings. The authors need to improve the paragraph they have added regarding the strengths of the manuscript:

- They can improve by telling how these findings help in clinical practice.

- Further emphasis on how these findings translate into improved prognosis and management of post traumatic brain injury patients.

- Additionally, they may also discuss briefly about how these findings help in improving the current guidelines and recommendations.

7. PLOS authors have the option to publish the peer review history of their article (what does this mean?). If published, this will include your full peer review and any attached files.

Reviewer #1: No

---

## [Author Response · Author response to Decision Letter 1]

17 Aug 2022

Thank you for the valued comments given by the reviewer. They are very much helpful in improving our manuscript. We have further revised the manuscript accordingly and wish to give details on each point of revision in the following statements. In the mark-up copy, the revision points are in red. The pages and paragraphs are also referred to those in the mark-up copy of the manuscript. Please be informed that the cleaned manuscript is for press and has no mark-up inside.

1. Regarding the strengths of the findings, the authors need to improve the paragraph they have added regarding the strengths of the manuscript:

- They can improve by telling how these findings help in clinical practice.

- Further emphasis on how these findings translate into improved prognosis and management of post traumatic brain injury patients.

- Additionally, they may also discuss briefly about how these findings help in improving the current guidelines and recommendations.

• Our response: We consider that all reviewer’s comments are relevant to the implications of our study findings for clinical practice and research. We, therefore, add a paragraph on this matter between the parts of limitations and Conclusions of the Discussion. In addition, we add a reference to support two sentences in the added paragraph.

• Page 21, Paragraph 5 – Page 22, Paragraph 1: 

The present results should be helpful for clinical practice and research in this area. Because treatment options for post-TBI behavioral symptoms remain unsatisfactory, as a promising intervention, BWLT may change the clinical practice and prognosis of patients with post-TBI depression and sleep disturbance. Clinicians may consider BWLT as a treatment option for post-TBI behavioral symptoms. In a recent review [55], guidelines suggested only neurobehavioral interventions and cognitive-behavioral therapy for rehabilitating behavioral and emotional disorders. The low to moderate certainty of our positive findings may impact future evidence-based practice guidelines and lead to a recommendation of BWLT for some post-TBI behavioral symptoms, e.g., depression, sleep disturbance. More trials in large sample sizes with longer study duration remain needed. Two reasons appear to support that patients with mild TBI should be a priority studied group. First, other than the benefits for depression and sleep disturbance, our subgroup analysis also found that BWLT might improve sleepiness in patients with mild TBI. Second, little has been known about the specific therapy for behavioral symptoms in mild TBI [55].

• Page 30, reference 55:

55. Lee SY, Amatya B, Judson R, Truesdale M, Reinhardt JD, Uddin T, et al. Clinical practice guidelines for rehabilitation in traumatic brain injury: a critical appraisal. Brain Inj. 2019;33: 1263–1271. doi:10.1080/02699052.2019.1641747

---

## [Editor Report · Decision Letter 2]

22 Aug 2022

Efficacy and acceptability of blue-wavelength light therapy for post-TBI behavioral symptoms: a systematic review and meta-analysis of randomized controlled trials

PONE-D-22-16609R2

Dear Dr. Srisurapanont,

We’re pleased to inform you that your manuscript has been judged scientifically suitable for publication and will be formally accepted for publication once it meets all outstanding technical requirements.

Kind regards,

Tariq Jamal Siddiqi

Academic Editor

PLOS ONE
---

## [Editor Report · Acceptance letter]

24 Aug 2022

PONE-D-22-16609R2 

Efficacy and acceptability of blue-wavelength light therapy for post-TBI behavioral symptoms: a systematic review and meta-analysis of randomized controlled trials 

Dear Dr. Srisurapanont:

I'm pleased to inform you that your manuscript has been deemed suitable for publication in PLOS ONE. Congratulations! Your manuscript is now with our production department. 

Kind regards, 

on behalf of

Dr. Tariq Jamal Siddiqi 

Academic Editor

PLOS ONE